# The Application of a Topology Optimization Algorithm Based on the Kriging Surrogate Model in the Mirror Design and Optimization of an Aerial Camera

**DOI:** 10.3390/s23167236

**Published:** 2023-08-17

**Authors:** Yubo Zhao, Lei Li

**Affiliations:** Changchun Institute of Optics, Fine Mechanics and Physics, Chinese Academy of Sciences, Changchun 130033, China; lilei_heu@hotmail.com

**Keywords:** primary mirror, optimized design, lightweight structure, finite element simulation (FES), Kriging surrogate model

## Abstract

In a Cassegrain optical system, the surface precision of the primary mirror is an important factor in the quality of the image. The design of a lightweight primary mirror with a high-quality optical surface is crucial. In this thesis, an integrated mirror light engine design optimization process is proposed for an aviation optoelectronic device. It is based on the Kriging surrogate model and nests the topology optimization algorithm, which constructs the mirror RMS value response surface and obtains the dominant relationship between mirror structure and surface accuracy. The optimal surface figure lightweight structure of the mirror is obtained by optimizing the surrogate model with an additive criterion and multi-objective optimization analysis. The root mean square value (RMS) of the corresponding primary mirror is 10.41 nm, which is better than 1/40 λ (λ = 632.8 nm). This meets the optical design specifications. The optimal primary mirror structure is analyzed by using the finite element method, which verifies the precision of the Kriging surrogate model. It has an error of 0.28%. The kinetic analysis of the primary mirror shows that the primary mirror does not yield to plastic deformation or even failure under a three-way 20 g acceleration load. This meets the environmental suitability requirements.

## 1. Introduction

The aerial camera has the advantages of being flexible, time-sensitive, highly accurate and high-resolution. It is an important technical tool in the acquisition of target information and strategic intelligence. With the increase in reconnaissance distance demand, the aerial camera optical system is developing towards high resolution, large field of view, large aperture, long focal length and multi-spectral band. Large-aperture mirrors have become the focus of research by scholars [1]. The increase in the aperture brings many difficulties to the structural design [2,3,4]. Firstly, it leads to an increase in the weight of the whole system structure, thus increasing the development cost. Secondly, gravity, temperature and other factors can cause large deformation of the mirror structure, which affects the surface precision of the optical surface and reduces the imaging quality of the system. Therefore, we need to solve the problem of how to design the mirror structure in a reasonable way while satisfying the environmental adaptability, as well as ensuring the lightweight and high surface precision of the mirror.

The topology optimization method can be used to find the optimal material distribution in the design domain under certain constraints. This is a powerful tool for structural design, which is used to obtain new configurations of structures [5]. It has important applications in the fields of aerospace [6,7], automotive [8,9], mechanical [10,11] and material design. Park [12,13] et al. analyzed the three-dimensional solid mirror model according to its subjection to polishing pressure loading and self-weight and designed the variables as solid unit density. They used the topology optimization method to design the first lightweight mirror with a 78% light-weighting rate. Lee [14] et al. subdivided the mirror into a structure consisting of many small units and optimized the design of the back reinforcement layout of the mirror. However, the design process did not consider the effect of gravitational deformation factors. Hu [15] et al. used the topology optimization design method to structurally optimize the flexible structure. They took the maximum first-order modal frequency as the constraint and the minimum precision value (PV) of the mirror as the objective function. Then, they extracted features based on the optimization results and modelled them, before, finally, obtaining the detailed shape parameters. Through this analysis, it was found that the mirror-back structure configuration was the key to improving the mirror face-shape precision. Compared to conventional lightweight configurations, topology optimization can provide higher lightweight rates and face shape precision. However, existing topology optimization methods for mirrors rely on certain empirical formulas, i.e., selected basic configurations, which make it difficult to obtain theoretically optimal structures.

The surrogate model is also known as Response Surface Model. It was proposed by Box and Wilson in the 1950s [16]. As the research on the surrogate model progressed, the method gradually moved towards multi-objective optimization, multidisciplinary optimization [17], global optimization and uncertainty optimization [18]. The predictive performance of many surrogate models has been extensively investigated by domestic and international scholars [19,20], which shows that different models are suitable for different applications. The Kriging model shows good approximation ability for strong nonlinear relations, which is a spatial interpolation technique based on stochastic process theory [21]. The ability to give unique approximation error estimates while solving engineering applications is the most significant feature of the Kriging model, which also shows strong adaptive capabilities when combined with adaptive optimization techniques [22]. Joseph et al. [23] proposed the blind-Kriging model using a combination of Bayesian feature selection techniques and experimental data. For simulation models that provide information on the gradient of the simulated response, the literature [24] summarizes the gradient-enhanced Kriging model that uses gradient information to improve the prediction precision of the Kriging model.

This thesis proposes an optimization method based on a combination of the Kriging surrogate model and topology optimization and applies it to the design of aerial camera mirror structures. In this way, the correspondence between the structural parameters of the mirror and its surface accuracy can be clearly characterized. The surrogate model idea is used to find the optimal configuration of the mirror under certain constraints, i.e., to meet the goal of lightweight and to obtain the optimal surface precision, to ensure optimal imaging quality of the optical system. There is little research on the joint design of Kriging surrogate models and topology optimization.

## 2. Initial Structure Model of Primary Mirror

As the core optical element of the Cassegrain optical system, the surface precision of the primary mirror plays a crucial role in the image quality of the camera. The primary mirror studied in this thesis is circular in shape. It has a diameter of 284 mm and a central light aperture of 75 mm. In order to design a mirror that is lightweight, is small in size and has excellent surface figure performance, the initial structure can be considered in terms of materials, basic configuration choices, support methods and mirror body thickness. This is a process of research, coordination and trade-offs.

### 2.1. Selection of Preparation Materials

In the manufacture of mirror bodies, materials should be selected that are easy to process. In order to reduce the reflection of the wavefront generated by a variety of aberrations, mirror materials need to have good physical structure stability to overcome the process of processing, assembly and transport to experience complex load conditions in order to ensure good surface shape precision. Mirror materials should also have high stiffness and good thermal stability. The physical properties of commonly used materials are shown in Table 1.

The mirror was made from aluminum alloy. The specific grade was 6061-T6, which is widely used in aerospace applications because of its ease of manufacture and molding, as well as its low cost. In optical systems, the use of aluminum alloy for both the mirror and its support structure allows for integrated heat dissipation and heat-free design, reducing the sensitivity of the structure to component materials.

### 2.2. Lightweight Configuration of the Mirror

The aluminum alloy material has the advantages of low cost and easy processing, but the material is dense and too heavy for a solid mirror with a diameter of 284 mm. Thus, it must be designed for lightweight, to reduce the weight of the whole machine. The lightweight design not only reduces the thermal equilibrium time but also improves the mirror’s surface precision and increases the mirror’s first-order intrinsic frequency.

The open-back configuration is widely used as the most common configuration for aerial camera mirrors. The design of the mirror-back matrix is one of the most effective ways to improve the mirror’s surface precision and lightness. In order to achieve the goal of increasing the weight reduction rate, the conventional open-back configuration uses a large number of weight-reducing holes in the mirror back.

### 2.3. Thickness of the Mirror

Precision by mirror body thickness is very influential. A reasonable selection of mirror body thickness can improve the primary mirror surface shape precision. The circular mirror diameter–thickness ratio and mirror surface self-weight deformation of the empirical formula is as follows:(1)$$\delta =\frac{3\rho ga^{4}}{16Et^{2}}=\frac{3\rho g{\left(D/t\right)}^{2}D^{2}}{256E}$$

In the above Equation, δ represents the PV value of the deformation of the mirror surface, g is the acceleration due to gravity and ρ is the material density of the mirror body. The radius and diameter of the mirror are represented by *a* and *D*, respectively. *t* is the thickness of the mirror and *E* is the modulus of elasticity of the material. The final thickness of the mirror was determined to be 50 mm by taking into account the above Equation and the overall dimensional constraints.

### 2.4. Theoretical Design of Support Point and Support Position

The support parameter of the mirror is the covariate that has the greatest impact on the accuracy of the topography. The support position and support interval are usually determined by Hall’s empirical formula, which is presented below:(2)$$N=\frac{1.5r^{2}}{t}\sqrt{\frac{\rho g}{E\delta }}$$

In the above Equation, *r* is the radius of the mirror body of the aerial camera, ρ is the mirror material density, g is the acceleration of gravity, *t* is the thickness of the large diameter mirror body, *N* is the minimum number of support points, *E* is the mirror material modulus of elasticity and δ is the maximum deformation of the mirror by its own weight. The radius of the distribution position can be determined according to the empirical formula given by Hindle:(3)$$R=\left(\frac{\sqrt{3}}{6}\right)D_{m}=0.28868D_{m}$$

In the above equation, R is the radius of the circumference where the support hole is located. $D_{m}$ is the maximum hole diameter of the mirror.

Based on the results of the above analysis, the final determination of mirror material was aluminum alloy. The mirror size was 284 mm, the central light aperture was 75 mm and the mirror body thickness was 50 mm. The support method selected was back six-point support. The sides of the mirror were manufactured using wire-cutting technology to create a flexible structure, which was used to reduce the effect of static forces on the mirror’s surface precision. The back of the mirror had three rectangular slots, the end of which was the position of the support point. The radius of the support was 85 mm.

## 3. Ultra-Lightweight Optimized Design of Mirror Primary

### 3.1. Design Parameters and Indicators

The main design parameters and indicators of the mirror were as follows:
Mirror size: 284 mm;Central light transmission aperture: 75 mm;Lightweight rate: >50%;Surface precision: RMS ≤λ/40 (λ = 632.8 nm) under 1 G gravity in the horizontal orientation of the optical axis;Mirror’s first-order inherent frequency: >600 Hz.

### 3.2. Topology Optimization Method for Mirror

After determining the main dimensions and basic configuration of the mirror, detailed mirror lightweight configurations were obtained in conjunction with traditional topology optimization methods, as explained in the following sections. The specific optimization flow chart is shown in Figure 1.

### 3.3. Topology Optimization Model for Mirror

The initial finite element model of the mirror was divided into a design domain and a non-design domain. The non-design domain was retained on the mesh, where the material could not be removed. It included the mirror connection interface, the back square slot, the part around the central vent and the mirror surface. For the design domain, the topology was optimized. The design domain and Non-design domains is shown in Figure 2.

### 3.4. Analysis of Optimization Conditions

The PV value and RMS value are the two core optical indicators for evaluating the surface precision of the mirror. When the PV value is smaller, the surface precision is better. In the mirror body modelling, the coordinate system was set as follows. The direction of the optical axis of the Cassegrain optical system was set to the *Z*-axis direction and the radial direction along the mirror was set to the *X*-axis and *Y*-axis directions.

In the practical operation of optoelectronic devices, the mirror is affected differently by gravity in different attitudes. According to design experience, when the mirror is affected by different directions of 1 G gravity conditions, the mirror surface deformation values of X and Y (i.e., two-directional sensitivity) are low and the Z-direction is more sensitive. Thus, in the topology optimization design process, if the mirror surface figure value meets the design requirements under 1 G gravity conditions in the Z-direction, the X- and Y-directions will naturally meet. Therefore, the surface shape precision value was PV ≤λ/10 (λ = 632.8 nm) and the minimum volume of the mirror in the Z-direction under the 1 G gravity condition was used as the boundary conditions for optimization. The structural stiffness of the mirror was used as the optimization objective for the topology optimization design of the mirror.

### 3.5. Optimized Design of Mirror Body

In the topology optimization process, the actual machining process was considered with manufacturing constraints and symmetry extraction constraints. The surface shape precision PV value and the minimum volume of the mirror were used as constraints; the minimum structural flexibility of the mirror was used as the objective function. Therefore, based on the SIMP method, the topology optimization was carried out using the solver in ABAQUS software. Finally, the conceptual configuration was extracted, using ANSA software to view the iterative optimization process of the mirror and the final optimization results, in order to carry out the detailed design. The optimization mathematical model is as follows:(4)$$\begin{array}{c} Findx=\left(x_{1},x_{2},\ldots x_{n}\right)\\ MinF_{c}\left(x\right)=F^{T}U=U^{T}KU=\overset{n}{\underset{i=1}{\sum }}f_{i}\left(x_{i}\right)u_{i}^{T}k_{i}u_{i}\\ S.t.\{\begin{array}{c} V=\overset{n}{\underset{i=1}{\sum }}x_{i}v_{i}\leq fV_{0}=V^{*}\\ F=KU\\ PV=max\left\{U_{1},U_{2},\ldots U_{N_{S}}\right\}-min\left\{U_{1},U_{2},\ldots U_{N_{S}}\right\}\leq PV^{*}=\lambda /10\\ 0\leq x_{i}{\left(i=1,2,\ldots ,n()\right)}_{min} \end{array} \end{array}$$

In the equation, the objective function is represented by $F_{c}$. The mirror external load is represented by *F*, i.e., 1 G gravity along the Z-axis. The volume variable is represented by V. The mirror initial structural volume and volume minima are represented by $V_{0}$ and $V^{*}$, respectively. PV and $PV^{*}$ are the initial peak and valley values of surface accuracy as well as the peak and valley minima of surface accuracy in the constraints, respectively. After iterative calculation, the topology optimization process of the mirror and the iterative curve of the objective function are as shown in Figure 3 and Figure 4.

The variable density method (SIMP, Solid Isotropic Material with Penalization Model) was used for the topology optimization of the aerial camera’s mirror. The finite element idea was used to consider the network cells of the mirror refinement as having a variable density. After topology optimization, the “variable density” values of all cells were classified as 0 or 1. Cells with a “variable density” value of 1 were retained. Similarly, the cells with a “variable density” value of 0 were removed from the mirror body, thus achieving a lighter mirror.

The following observations can be seen from the figure:In the results of the topology optimization of the mirror body based on the maximization of stiffness, the material in the design domain was concentrated around the support points and a lightness rate of 55% was achieved;Triangular and fan-shaped reinforcement bars were used around the six support points to improve the local stiffness in their vicinity;The reinforcement needed to be designed to retain sufficient thickness to increase its supporting stiffness.

In this section, a topology optimization model was developed based on the basic structure of the mirror. It was verified that the topology-optimized mirror had a significant improvement in lightness and surface precision compared to the conventional design. In a competitive world, the philosophy of structural design should be that, if a solution is feasible, it is to be made as good as possible. For aerial camera mirrors, the surface precision directly affects the imaging quality of the optical system. It is clearly impractical to perform finite element calculations for all support positions and support intervals within a certain range. This thesis proposes an optimization method that nests the Kriging surrogate model and topology optimization to find the optimal structural design of the mirror under certain constraints.

## 4. Research on the Topology Optimization Algorithm Based on the Kriging Surrogate Model

For engineering applications, most structural optimization is carried out by means of finite element analysis, which can be a great challenge when the problem is complex and the number of objectives to be optimized is large. Many scholars have brought the theory of the surrogate model to engineering structural optimization problems in order to reduce the computational time cost of the model. For the aerial camera mirror, there is a strong non-linear relationship between the different initial design variables and the surface precision. This section uses the surrogate model approach to convert the problem into a global optimal search, avoiding the complex non-linear relationship of the original system.

### 4.1. Surrogate Model Theory

The advantage of the surrogate model approach is that, instead of calculating the performance metrics for all sample points in the design domain, the known sample points are used to predict the performance metrics for the unknown sample points. Firstly, the sample points were obtained by Latin hypercube sampling in the design domain as input. The performance metrics of the sample points were obtained by simulation or physical experimentation as output. Secondly, the response relationship between the design variables and the performance indicators was fitted using the surrogate model. Then, based on the approximate response relationship, a multi-objective optimization algorithm was used to obtain the optimal solution set. Since the number of sample points was small and did not represent the characteristics of all samples, it was difficult to obtain the response surface of the relationship between design variables and performance indicators with high precision at one time. Therefore, a model update strategy was subsequently required to improve the precision of the surrogate model.

Among the many surrogate models, the first and most widely used is the polynomial Response Surface Model. The Kriging surrogate model as its representative is a typical interpolation class surrogate model with good non-linear fitting ability, which is widely used at present.

In summary, the basic idea of the Kriging surrogate model-based topology optimization algorithm for solving the mirror optimal surface figure is as follows. Firstly, the sample points are extracted from the design domain constructed by mirror support position and support interval by Latin hypercube sampling method. The optimal topology of a single sample point is obtained according to the topology optimization method in the previous section. Then, the RMS response of the primary mirror under this structure is obtained according to the surface figure analysis method of the light engine. Through this process, a set of mirror structure samples and their corresponding response values are obtained. This process is repeated to obtain the response values for the entire sample set. The Kriging surrogate model is then fitted to the test samples to obtain a response surface that meets the accuracy requirements. The flow of topology optimization based on the Kriging surrogate model is shown in Figure 5.

### 4.2. Kriging Surrogate Model

The Kriging surrogate model stochastic function corresponding to the response *Z(x)* at any point *x* in the sample space can be expressed as follows:(5)$$Z\left(\vec{x}\right)=b^{T}\left(\vec{x}\right)\cdot \vec{\beta }+\varepsilon \left(\vec{x}\right)$$

In Equation (5), $Z\left(\vec{x}\right)$ is the model response, $b^{T}\left(\vec{x}\right)$ is the regression function used for estimation, $\vec{\beta }$ is the regression coefficient, $b^{T}\left(\vec{x}\right)\cdot \vec{\beta }$ is the mean square error of the random function $Z\left(\vec{x}\right)$ and $\varepsilon \left(\vec{x}\right)$ is the random error. The covariance matrix of $\varepsilon \left(\vec{x}\right)$ is as follows:(6)$$cov[\varepsilon \left(x_{i}\right),\varepsilon \left(x_{j}\right)]=\sigma^{2}R\left[\varepsilon \left(x_{i}\right),\varepsilon \left(x_{j}\right)\right]$$

σ is the standard deviation of the random process and $R\left[\varepsilon \left(x_{i}\right),\varepsilon \left(x_{j}\right)\right]$ is the correlation function between the i-th sample and the j-th sample.

The covariance functions commonly used in Kriging models are as follows:
Linear covariance function (7)$$Cov\left(Z\left(x_{i}\right),Z\left(x_{j}\right)\right)=\{\begin{array}{l} C_{0}\left(1-\frac{h}{a}\right),h\leq a\\ 0,h>a \end{array};$$Three-time curve covariance function (8)$$Cov\left(Z\left(x_{i}\right),Z\left(x_{j}\right)\right)=\{\begin{array}{c} C_{0}\left(1-7\frac{h^{2}}{a^{2}}+8.75\frac{h^{3}}{a^{3}}-3.5\frac{h^{5}}{a^{5}}+0.75\frac{h^{7}}{a^{7}}\right),h\leq a\\ 0,h>a \end{array};$$Spherical covariance function
(9)$$Cov\left(Z\left(x_{i}\right),Z\left(x_{j}\right)\right)=\{\begin{array}{c} C_{0}\left(1-\frac{3h}{2a}+\frac{h^{3}}{2a^{3}}\right),h\leq a\\ 0,h>a \end{array};$$Power exponential covariance function
(10)$$Cov\left(Z\left(x_{i}\right),Z\left(x_{j}\right)\right)=C_{0}e^{\frac{h}{a}}.$$In the above covariance functions, $C_{0}$, h and a are all unknown parameters to be determined. Among the above classical covariance functions, linear covariance functions are mostly used for optimal solutions in one-dimensional space. The cubic and spherical covariance functions are mostly used for optimal solutions in less than three-dimensional space. The Gaussian covariance function, more suitable for complex multidimensional estimation, is chosen in this thesis;Gaussian covariance function
(11)$$R\left(\theta ,x_{i},x_{j}\right)=exp[-\sum_{m=1}^{M}\theta_{m}{\left(x_{i}-x_{j}\right)}^{p_{m}}].$$

The Gaussian covariance function optimizes the parameters of interest using maximum likelihood estimation, where $\theta_{m}$ represents the sensitivity of the m-th variable. It influences the components of the Gaussian covariance function in each direction. $p_{m}$ is the parameter characterizing the smoothness. In order to obtain the optimal covariance function parameter $\theta_{m}$, maximum likelihood estimation is used with a likelihood function as follows:(12)$$L\left(\theta ,\beta ,\sigma^{2}\right)=\frac{1}{{\left(2\pi \sigma^{2}\left(\theta \right)\right)}^{\frac{N}{2}}{\left|R\left(\theta \right)\right|}^{\frac{1}{2}}}e^{\frac{{\left(Z-B^{T}\beta \right)}^{T}R^{-1}\left(\theta \right)\left(Z-B^{T}\beta \right)}{2\sigma^{2}}}$$

R is the correlation matrix for all observed samples:(13)$$R=\left[\begin{array}{ccc} R\left(\theta ,x_{1},x_{1}\right) & \cdots & R\left(\theta ,x_{1},x_{N}\right)\\ \vdots & \ddots & \vdots \\ R\left(\theta ,x_{N},x_{1}\right) & \cdots & R\left(\theta ,x_{N},x_{N}\right) \end{array}\right]$$

The optimal covariance parameter $\theta_{m}$ is determined by solving for the maximum value of the likelihood function L to obtain the optimal covariance function. This transforms the essence of model building into the problem of selecting the optimal $\theta_{m}$ to minimize the random error $\varepsilon \left(\vec{x}\right)$ in Equation (5).

As the Kriging model is an interpolated surrogate model, the sample points generated by the experimental design and the refinement points generated during the update of the surrogate model all pass through the real response surface, i.e., the predicted values at the sample points and refinement points are equal to the values calculated by the simulation. This interpolation method can better predict the response area closer to the sample points or refinement points, while the sparse sample areas are difficult to predict. Therefore, the precision of the response surface cannot be evaluated by the existing sample points and validation points need to be added to evaluate the fitting precision of the surrogate model.

### 4.3. Kriging Surrogate Model for the Mirror Structure Design

The mirror’s back-support position and spacing parameter X=(l,x) were selected as input parameters. The mirror’s surface precision Y=y under the corresponding input parameters was used as the output response to build the Kriging surrogate model for the mirror’s structural design. m known support parameters were obtained by solving the physical model. The surface figure response $\vec{Y}=\vec{y}$ of the mirror for m known support parameters $\vec{X}=\left(\vec{l},\vec{x}\right)$ was obtained by solving the simulation for the physical model, among them,
(14)$$\vec{X}=\left[\begin{array}{c} X_{1}\\ X_{2}\\ \vdots \\ X_{m} \end{array}\right]=\left[\begin{array}{cc} l_{1} & x_{1}\\ l_{2} & x_{2}\\ \vdots & \vdots \\ l_{m} & x_{m} \end{array}\right],\;\vec{Y}=\left[\begin{array}{c} y_{1}\\ y_{2}\\ \vdots \\ y_{m} \end{array}\right]$$

Substituting Equation (14) into Equation (5) gives
(15)$$Y\left(\vec{x}\right)=b^{T}\left(\vec{x}\right)\cdot \vec{\beta }+\varepsilon \left(\vec{x}\right)$$

The Kriging surrogate model for the mirror can be obtained by means of Equation (15). Using the established model to estimate the response values at the unknown parameter, $\;X^{*}=\left(l^{*},x^{*}\right)$ and then
(16)$$Y^{*}\left(X^{*}\right)=\overset{1}{\underset{m}{\sum }}w_{i}\left(X\right)Y\left(X_{i}\right)$$

$w_{i}$ is the weight corresponding to each known parameter, to ensure the unbiasedness and optimality of Equations (6), (11), (15) and (16). The unbiased and optimal nature of the Kriging surrogate model allows the corresponding weighting coefficients $w_{i}$ to be derived.

Once the Kriging surrogate model has been built, metrics are needed to evaluate the performance of the Kriging surrogate model for the mirror. The mean squared error MSE is generally used for evaluation:(17)$$MSE\left(X^{*}\right)=\frac{1}{M}\left(\overset{1}{\underset{m}{\sum }}{\left(Y^{*}-Y\right)}^{2}\right)$$

In the equation, Y and $Y^{*}$ are the exact value of the function and the Kriging model prediction, respectively. In this thesis, the dimensionless mean squared error was chosen, while the relative mean squared error RMSE was chosen to evaluate the performance of the mirror rotor Kriging surrogate model:(18)$$RMSE\left(X^{*}\right)=\frac{1}{Y^{*}{}^{2}M}\left(\overset{1}{\underset{m}{\sum }}{\left(Y^{*}-Y\right)}^{2}\right)$$

### 4.4. Latin Hypercube Sampling

A small selection of samples within the mirror’s support parameter sample space was used as the original sample for building the Kriging model for the optimal design of the mirror’s structure. The accuracy of the subsequent Kriging surrogate model of the mirror was strongly influenced by the original sample. Therefore, the selected samples were required to be representative, as well as randomly and uniformly distributed throughout the sample space. Considering the high computational cost per sample, the number of samples selected had to be kept within a certain range. In this thesis, 48 samples were selected in the support parameter sample space as the original samples for building the Kriging surrogate model. The support position ranged from 71 mm to 99 mm, accounting for 50% to 70% of the maximum radius of the mirror. The support interval ranged from 41 mm to 85 mm, taking into account the assembly constraints of the whole machine and the machining process. The Latin hypercube sampling is shown in Figure 6.

Similarly, Latin hypercube sampling was used to select 14 samples to be tested that differed from the 48 original samples used to validate the precision of the mirror’s Kriging model.

### 4.5. Update Strategy of Surrogate Model

In order to improve the response relationship between the surrogate model fitted inputs and outputs, an update strategy was provided to add new sample points based on the current model construction information to facilitate the subsequent acquisition of the optimal solution.

Common Addition Guidelines:
MP Addition Guideline

The minimize prediction (MP) addition guideline was used to add the optimal solution of each optimization to the surrogate model as a new sample point. The advantage of MP Addition Guidelines is that convergence is very fast. It is widely used in multi-objective optimization as it saves a lot of time in optimization. The disadvantage is that it can fall into local optima;
2.MSE Addition Guideline

The mean square error (MSE) addition guideline chooses to add points directly to the focus area, i.e., the maximum predicted mean square error. As such, it was convenient to use the MSE pointing criterion. The advantage of the MSE criterion over the MP criterion is that it improves the precision of the global surrogate model but at the expense of some computational time.

Of the above-mentioned addition guidelines, the MP addition guideline is the most general, which can be easily extended to multi-objective optimization designs. The Kriging model provides a mean square error estimate, so the MSE addition guideline is more applicable to the Kriging surrogate model.

### 4.6. Precision Evaluation Method of Kriging Surrogate Model

The method of evaluating the precision of the Kriging surrogate model is an inverse problem of finding the cause from the effect. According to the previous section, 14 samples to be tested were selected by Latin hypercube sampling. The Kriging surrogate model of the mirror was established by using the original samples to obtain the response surface corresponding to all the samples in the sample space. The response surface corresponding to all the samples in the sample space was obtained. The response values of the 14 samples to be tested were input. The points on the response surface closest to the response value of the point to be tested were found. The support parameter corresponding to the response value to be tested was identified. In addition, the problem was transformed into an optimal search in space. If the identification was successful, the precision of the surrogate model was considered to meet the design requirements. If not, the original number of sample points of the surrogate model was added through the MSE addition criterion to optimize the surrogate model:(19)$$\begin{array}{c} Rank{∥Y^{*}-Y∥}_{2}\\ \min{∥Y^{*}-Y∥}_{2} \end{array}$$

In Equation (19), the difference between the predicted and target values is used to find the length of the vector in Euclidean space. The support parameter for the smallest two-parameter option is found to be the result of the support parameter identified by the Kriging surrogate model. Once the surrogate model has been constructed, the method can also be used as a guide for structural design.

### 4.7. Construction and Optimization of the Kriging Surrogate Model for the Mirror

As explained in the previous content, the support position and support interval of the mirror were selected as random parameters. After determining the optimal topology according to the different back-support parameters, a mirror surface figure was calculated to obtain the response values and a Kriging surrogate model of the mirror was constructed. The relative mean square error of the surrogate model is shown in the figure below.

The whole sample space response was predicted by the surrogate model. After obtaining the sample spatial response surface, the mean square error MSE and relative mean square error RMSE for each point of the response surface were found using Equation (17). As can be seen from Figure 7, the MSE and RMSE values at the edge of the support parameter design domain of the mirror are larger. This means that this Kriging model does not perform well to the edges. However, it has better diagnostic prediction performance for sample points in the middle of the design domain. The performance of the Kriging surrogate model is enhanced by increasing the initial samples at the edges of the sample space, according to the MSE addition guideline described above.

The mirror’s surrogate model had a significant reduction in MSE and RMSE values in the sample space after adding the original samples at the boundary, with the maximum MSE value reduced from 80 to 3 and the maximum RMSE value reduced from 0.6 to 0.01. This means that the performance of the surrogate model improved significantly after adding the original samples.

### 4.8. Computer Test Verification

In the previous section, the mean squared error MSE and the relative mean squared error RMSE were used as indicators to assess the performance of the surrogate model. In this section, the accuracy of the Kriging surrogate model for the mirror is assessed through practical validation of the sample. Similarly to before, by Latin hypercube sampling, 14 samples to be tested were randomly and uniformly selected within the sample space that is different from the original samples. The response values of the samples to be measured were obtained by simulation using the finite element method. The supported parameter was identified using Equation (19), which searches for the minimum distance between the value to be measured and the response surface derived from the surrogate model and the corresponding sample input parameter was the identified support parameter.

The mirror’s Kriging surrogate model was utilized to identify the support parameter to be measured and the results are shown in Table 2. By adjusting the initial θ in Equation (12), the model was automatically optimized to obtain the best identification result under $\theta^{\prime }=\left[2.025,\;2.399\right]$. The assessment results of the surrogate model are shown in Table 3. The support position recognition precision was 75% and the support interval recognition precision was 93%. The overall probability of accurate support parameter recognition was over 72%.

The samples to be tested that were not identified accurately were (82,65), (93,61), (96,45) and (85,45). According to Figure 8, the Kriging surrogate model of the created mirror had a large RMSE at all the above locations. Therefore, the surrogate model do not have a high prediction precision. Although the recognition precision was only 72%, the difference between the recognition result and the true value of the sample to be tested (the actual support position and support interval) was small in the samples to be tested that were not identified accurately. The incorrectly identified support parameter was found to be at the second- or third-smallest distance (when the smallest value obtained from ${∥Y^{*}-Y∥}_{2}$ was the same as the support parameter to be measured and the parameter was correctly predicted), as shown by the ranking of the distance between the measured value and the predicted value of the surrogate model ${∥Y^{*}-Y∥}_{2}$ from smallest to largest.

In order to improve the prediction precision, it was decided to increase the original sample to further optimize the Kriging surrogate model of the mirror. However, in view of the actual cost of expanding the original sample (the finite element solution is time-consuming and computationally intensive), it was decided to add the accurately identified samples to the original sample space to form a new sample space, thus building up the first optimized Kriging surrogate model of the mirror. After the second identification, the accurately identified samples were added to the original sample space and the surrogate model was optimized until the identification of all the support parameters was completed (or the remaining samples could not be identified accurately).

The whole identification process was optimized twice by adding the samples with the correct support position and interval to the original sample space to build a new sample space, thus obtaining a better-performing surrogate model. This method also ensured that the surrogate model had a certain degree of heritability while updating the surrogate model.

The support parameters of the mirror were identified and the results are shown in Table 4. Before the first model optimization, the Kriging surrogate model for the initial generation of mirror accurately identified 10 sets of support parameters. After the first model optimization, the second-generation Kriging surrogate model accurately identified three sets of support parameters. After the second model optimization, the accuracy of the Kriging surrogate model of the third-generation mirror was further improved. The assessment results of the surrogate model are shown in Table 5. The accuracy of support position recognition was over 92%. The accuracy of support interval recognition was 100%. The overall probability of accurate support parameter identification exceeded 93%.

After optimizing the Kriging surrogate model twice, the number of samples not identified accurately was (96,45) and the identification result was (97,45), which was closer to the prediction of (98,45) than the initial Kriging model. This represents an improvement in the overall precision of the model, with a 1 mm difference in support interval. The impact on the surface precision was negligible. The overall probability of accurate support parameter identification was over 93%. Overall, the accuracy of the support parameter identification was improved by optimizing the surrogate model several times. The accuracy of the developed mirror surrogate model meets the requirements of finding the optimal surface figure for the actual structural design and guiding the design.

### 4.9. Determination of the Parameters of the Optimal Structure of the Mirror

In the previous thesis, a Kriging surrogate model was developed for the two target variables of mirror back-support position and support interval. The model was optimized using the Addition Guidelines and the accuracy of the model was verified. A high-accuracy surrogate model of the mirror was obtained. By using the response surface of the surrogate model, we could finally obtain the topology optimization structure of the support parameter corresponding to the optimal surface figure of the mirror. The response surface is shown in Figure 9.

From the RMS value of the mirror given by the response surface of the surrogate model, we can see that the support parameter of the optimal surface figure was (95,85), which corresponded to a surface figure RMS value of 10.41. Combined with the topology optimization process based on the Kriging surrogate model, we could finally obtain the mirror structure of the theoretical optimal aerial camera, as shown in Figure 10. According to the topology optimization flow chart, the lightweight structure of the mirror is mainly open in the back of the mirror to open the fan and triangular lightweight holes, with three rectangular grooves to manufacture flexible structures; the rectangular groove and the center hole are not subject to force to select a large-size sector hole, a small size sector lightweight hole is set between the six support points and the edge of the mirror is made of stiffeners to ensure the rigidity of the mirror.

## 5. Thermal Integration Analysis of Mirror Light Engine for Aerial Camera

The topology optimization algorithm of the Kriging surrogate model was used to obtain a structural model of the mirror, but further finite element analysis and environmental suitability testing of the component structure were required. On the one hand, finite element static and dynamic analysis was used to determine whether the mirror could meet the environmental suitability requirements under complex operating conditions. On the other hand, the primary mirror RMS values given by the algorithm were further verified by means of finite element surface figure testing.

### 5.1. Mirror Hydrostatic Analysis

The mirror is not consistent in the direction of the optical axis during operation, which is different from the structural optimization design. In the static analysis of the mirror, it is necessary to apply the load in the X-, Y- and Z-directions in order to fully reflect the deformation of the mirror surface and to evaluate its performance. The mirror finite element model was established using ANSA software. The constraints were set to full degrees of freedom and applied to the screw connections to simulate the actual environmental conditions. ABAQUS software was used to carry out the finite element analysis of the mirror under the three load conditions. The optical surface deformation cloud picture is shown in Figure 11. The mirror and its support structure were also made of aluminum alloy to allow for an integrated thermal dissipation and non-thermal design, so that thermal analysis was not simulated separately.

### 5.2. Mirror Surface Figure

The mirror optical surfaces are subject to changes in gravity and other mechanical loading environments that affect the final image quality. Two core optical indicators, PV value and RMS, are often used to evaluate mirror surface deformation. The difference between the peak and trough values of optical surface distortion is expressed as the PV value:(20)$$\mathrm{PV}=\Delta \delta min_{max}$$

The PV value alone does not give a true indication of the degree of superiority or inferiority of the optical surface performance. RMS is the root mean square of the difference between all nodes on the mirror-deformed surface and the fitted surface. Both need to be used together as an evaluation criterion:(21)$$\mathrm{RMS}=\sqrt{\frac{1}{N}\Sigma_{I}^{N}(\Delta \delta_{i}-\Delta \bar{\delta }{)}^{2}}$$

In the Equation (21), $\Delta \bar{\delta }$ is the average difference between all nodes on the deformed surface and the fitted surface. $\Delta \delta_{i}$ is the deviation value of the *i*-th node on the deformed surface. There are N nodes on the deformed surface. As the mirror surface deformation RMS value contains the deformation data of each node on the deformation surface, it contains more comprehensive information on the deformation data, which can truly and effectively reflect the actual performance of the mirror optical surface.

Matlab software has a powerful data processing function, which can quickly perform batch processing of the mirror surface deformation data after FEA. The deformation data were fitted by Zernike, using the least squares method in Matlab software. The final calculated PV and RMS values of the optical surface with the displacement of the rigid body removed are shown in Figure 12.

As can be seen from Table 6, the surface figure precision of the mirror met the design specifications (PV ≤ λ/10, RMS ≤ λ/40) as required in Section 2.1 of this thesis. The difference between the RMS value of the surface figure in the Z-direction of the mirror given by the finite element analysis and the predicted value of the optimal surface figure given by the Kriging surrogate model was 0.03 nm, with an error of 0.28%. This further confirms the superiority of the surrogate model method.

### 5.3. Analysis of Mirror Dynamics

Mirror components are subject to complex dynamics during operation, which can cause loosening or deformation of the structure or even structural failure. Therefore, in addition to the static analysis of mirror components, it was necessary to carry out the necessary dynamic analysis to determine whether the structure was subject to strength failure or resonance under dynamic conditions.

#### 5.3.1. Modal Analysis

Modal analysis is mainly used to study the inherent frequency of the mirror assembly and to analyze the vibration pattern of the assembly. It not only can determine whether the whole assembly exhibits resonance phenomena, but also can reflect the stiffness size of the structure, which is the basis of some other dynamic analyses. In the modal analysis of mirror components, the mirror component can usually be regarded as undergoing ideal undamped free vibration. Its vibration equation is expressed as follows:(22)$$\left[M\right]\left\{\ddot{u}\right\}+\left[K\right]\left\{u\right\}=\left\{0\right\}$$

For simple harmonic motion $\left\{u\right\}={\left\{\phi \right\}}_{i}cosw_{i}t$, then
(23)$$\left[K\right]{\left\{\phi \right\}}_{i}=w_{i}^{2}\left[M\right]{\left\{\phi \right\}}_{i},$$
namely,
(24)$$\left(\left[K\right]\right)-w_{i}^{2}\left[M\right]{\left\{\phi \right\}}_{i}=\left\{0\right\}$$

${\left\{\phi \right\}}_{i}$ exists a non-zero solution; therefore,
(25)$$\left|\left[K\right]-w_{i}^{2}\left[M\right]\right|=0$$

The calculated eigenvalues $w_{i}$ (*i* = 1,2,…N) are the intrinsic frequencies of the whole component system, which are then inserted into Equation (24) to calculate ${\left\{\phi \right\}}_{i}$ (*i* = 1,2,…N). This is the vibration pattern corresponding to the intrinsic frequencies.

Based on the above principles, the mirror finite element model was modeled by applying boundary conditions and setting the solution type. The first six orders of modal vibration, along with their specific values, are shown in Table 7 and Figure 13.

From the results of the modal analysis of the mirror, it can be seen that the first-order inherent frequency of the component structure was 1018.3 Hz. The dynamic stiffness can meet the design index requirements.

#### 5.3.2. Analysis of Mirror Dynamics

The aerial camera is subjected to prolonged, high-intensity acceleration loads during operation, which may cause large deformations and displacements of optical devices such as the mirror. In order to ensure that the primary mirror will not fail due to acceleration load, 20 g acceleration load was applied in the X-, Y- and Z-directions. The stress cloud pictures are shown in Figure 14.

The specific data and description of the stresses and displacements of the mirror assembly under acceleration loads are shown in Table 8.

As can be seen from the table data, the maximum stress in the mirror assembly under 20 g acceleration load in all three directions occurred at the lateral wire-cutting connection. The maximum value was 10.98 MPa, which is much lower than the yield stress of aluminum alloys. The maximum displacement is 4.6 μm, which occurred at the panel edge of the primary mirror under Z-directional overloading. Therefore, the primary mirror will not yield to plastic deformation or even failure under a three-way 20 g acceleration load.

## 6. Conclusions

The purpose of this paper is to optimize the design of the primary mirror in an aerial camera to obtain high surface shape accuracy, high stability and ultra-lightweight. On the basis of the basic structure of the mirror, a topology optimization model with the maximization of the mirror stiffness as the objective and the minimization of the volume fraction and PV value as the constraints is established. The lightweight and surface precision has been improved compared to the conventional design. The weight reduction rate is 55%. The support position and support interval, which have the greatest influence on the surface precision of the mirror, are selected as the optimization objectives. The root mean square RMS value of the mirror is used as the response to build the Kriging model. In order to improve computational efficiency, the method of optimizing the Kriging surrogate model has been proposed several times. After optimization three times, the model is evaluated. The results are as follows. The accuracy of support position recognition exceeds 92%, the accuracy of support interval recognition is 100% and the overall probability of accurate support parameter recognition exceeds 93%. The response surface of the Kriging surrogate model and the theoretical optimal configuration of the mirror are obtained for the high-accuracy aerial camera mirror. The corresponding optimal surface figure RMS value is 10.41 nm and the corresponding support parameter is (95,85). The surface figure RMS value is better than the surface figure RMS value, corresponding to the support parameter given by the traditional empirical formula. A light engine thermal integration analysis is performed on the optimal mirror structure. When the mirror is subjected to self-weight in the X-, Y- and Z-directions, the optical surface RMS values are 7.62 nm, 7.62 nm and 10.44 nm, respectively. This is 0.03 nm away from the 10.41 nm given by the algorithm, which is an error of 0.28%. This further validates the high accuracy of the constructed Kriging surrogate model. The results of the kinetic analysis of the mirror show that the intrinsic frequency of the mirror is much greater than that required by the design specifications. When subjected to impact loading, no yielding plastic deformation or even damage failure occurs under a three-way 20 g acceleration load. This paper proposes a new optomechanical design model that shows the relationship between mirror structure and surface accuracy. This provides an important reference value for the optimized design of the primary mirror of an aerial camera.

## Figures and Tables

**Figure 1 sensors-23-07236-f001:**
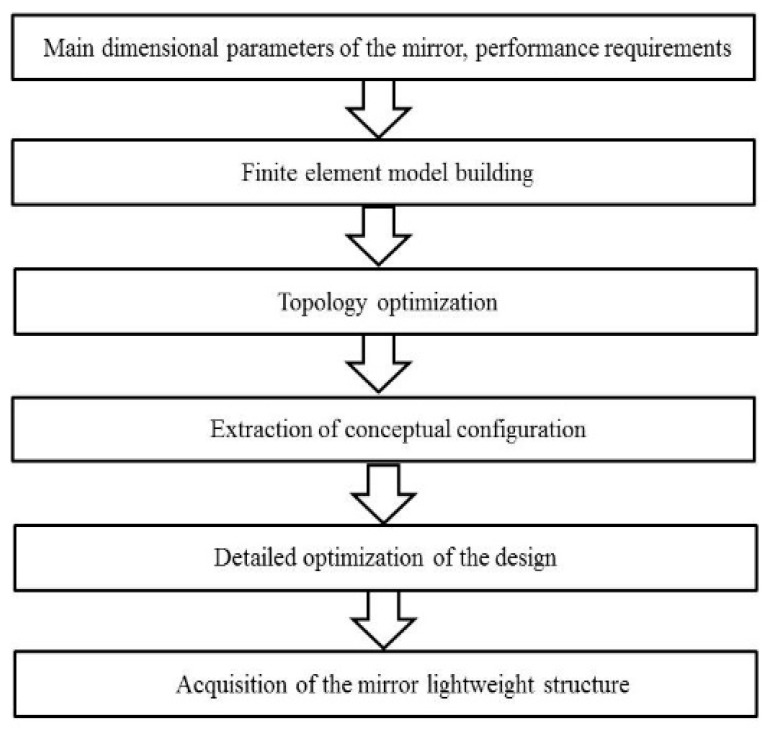
Flow chart of topology optimization.

**Figure 2 sensors-23-07236-f002:**
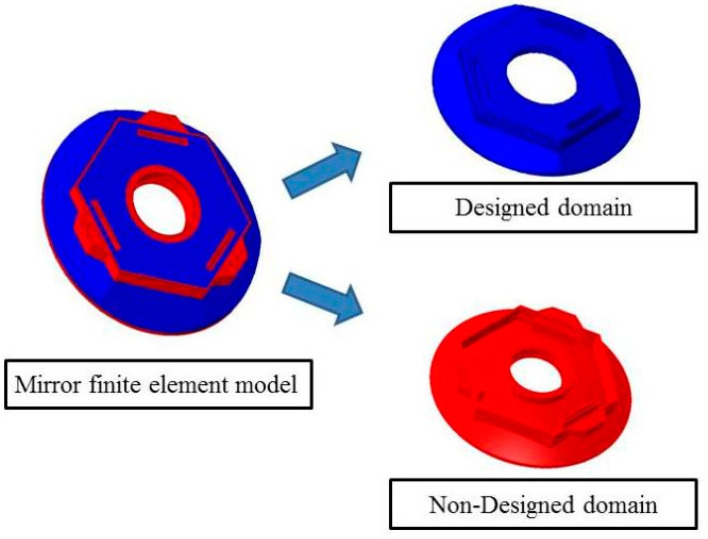
Mirror design domain and non-design domain.

**Figure 3 sensors-23-07236-f003:**
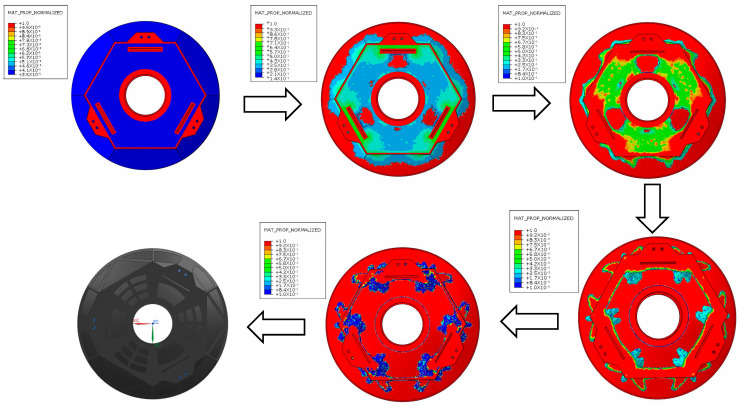
Process diagram for topology optimization of mirror.

**Figure 4 sensors-23-07236-f004:**
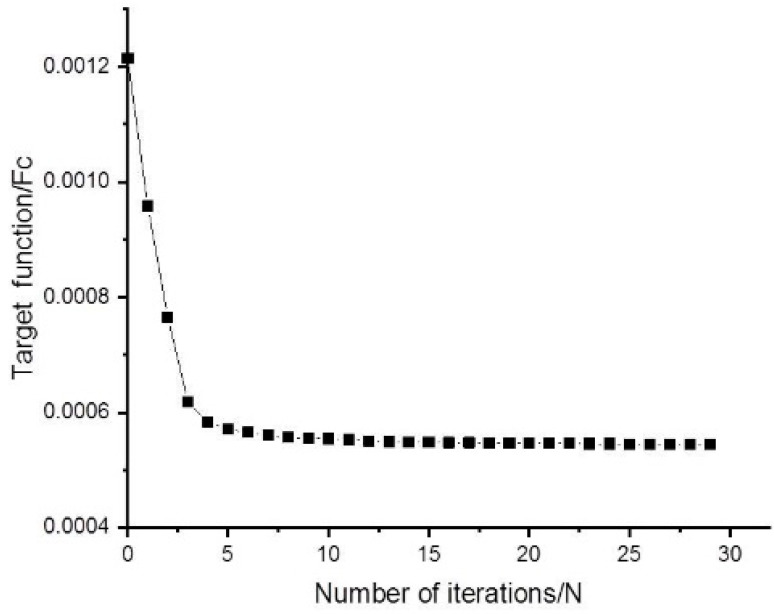
The iterative curve of the objective function.

**Figure 5 sensors-23-07236-f005:**
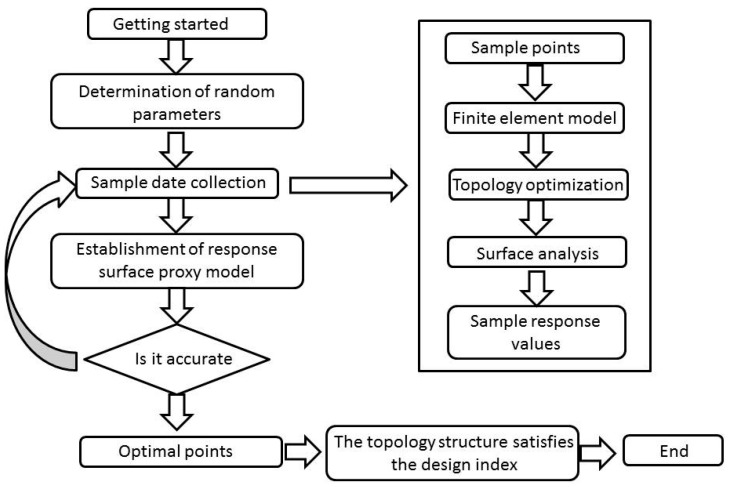
Topology optimization process based on Kriging surrogate model.

**Figure 6 sensors-23-07236-f006:**
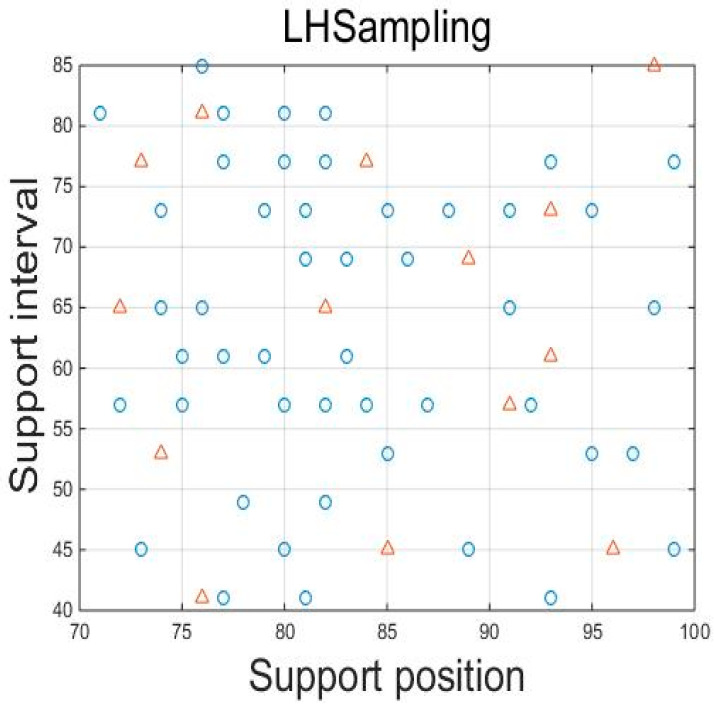
Latin hypercube sampling (blue circle—original samples; red tringle—Test samples).

**Figure 7 sensors-23-07236-f007:**
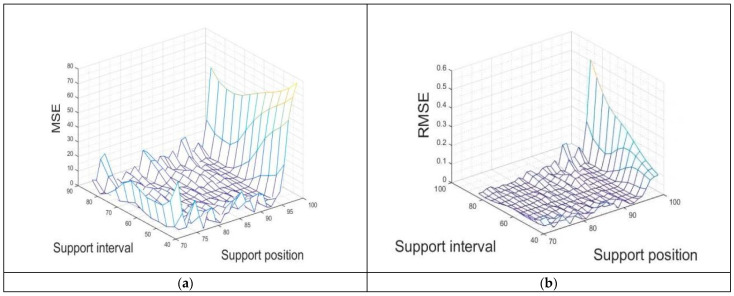
Mirror response surface. (**a**) Mean squared error values; (**b**) Relative mean square error values.

**Figure 8 sensors-23-07236-f008:**
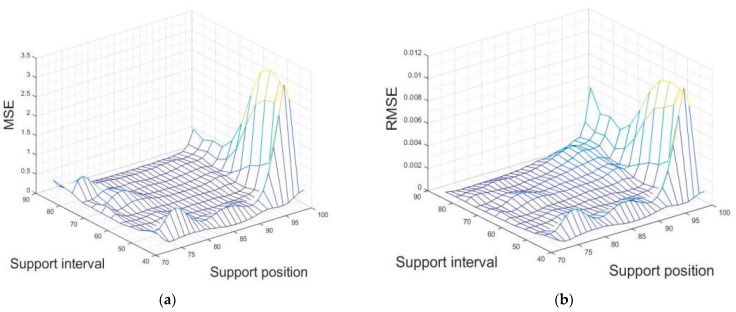
Mirror response surface after adding original samples. (**a**) Mean squared error values after adding original samples; (**b**) Relative mean square error values after adding original samples.

**Figure 9 sensors-23-07236-f009:**
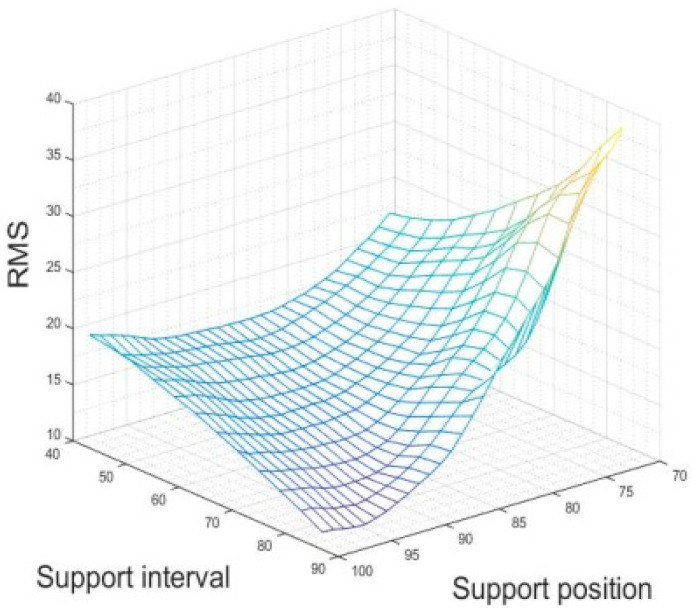
Response surface of the surrogate model with high-precision mirror.

**Figure 10 sensors-23-07236-f010:**
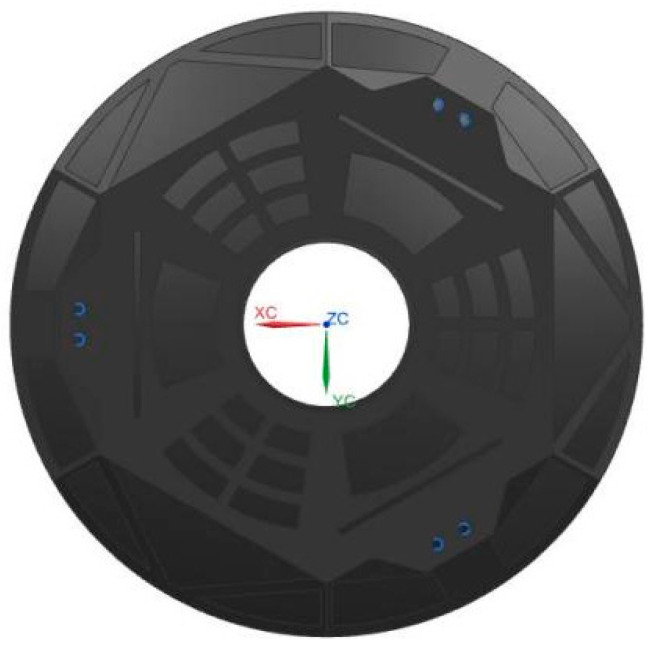
Structure of topology optimization of mirror optimal surface figure.

**Figure 11 sensors-23-07236-f011:**
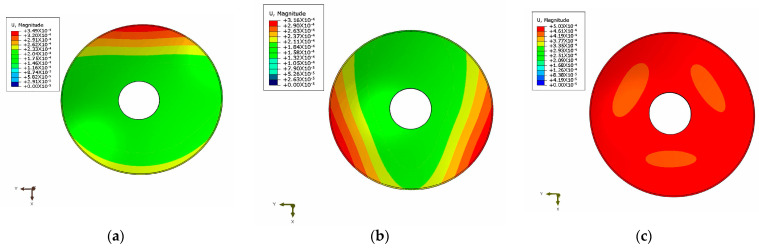
Cloud picture of the optical surface deformation of the mirror under gravity in the X-, Y- and Z-directions. (**a**) The X-direction is affected by gravity; (**b**) The Y-direction is affected by gravity; (**c**) The Z-direction is affected by gravity.

**Figure 12 sensors-23-07236-f012:**
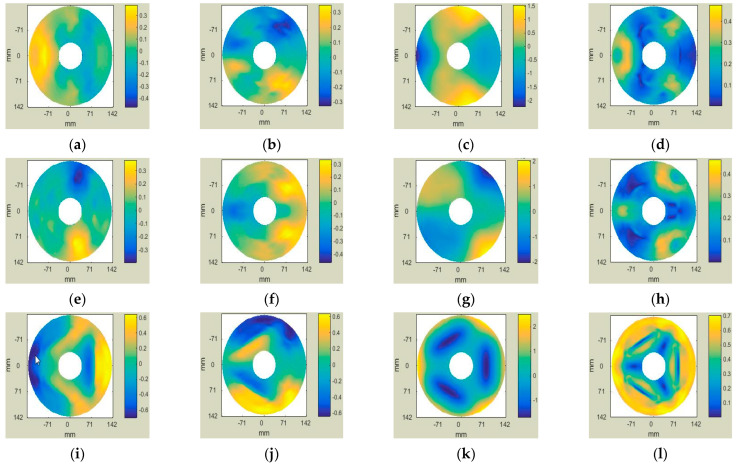
Optical surface for removal of rigid body displacement by gravity in the X-, Y- and Z-directions of the mirror. (**a**) Slope X Map (The X-direction is affected by gravity); (**b**) Slope Y Map (The X-direction is affected by gravity); (**c**) Surface Map (The X-direction is affected by gravity); (**d**) Slope Magnitude Map (The X-direction is affected by gravity); (**e**) Slope X Map (The Y-direction is affected by gravity); (**f**) Slope Y Map (The Y-direction is affected by gravity); (**g**) Surface Map (The Y-direction is affected by gravity); (**h**) Slope Magnitude Map (The Y-direction is affected by gravity); (**i**) Slope X Map (The Z-direction is affected by gravity); (**j**) Slope Y Map (The Z-direction is affected by gravity); (**k**) Surface Map (The Z-direction is affected by gravity); (**l**) Slope Magnitude Map (The Z-direction is affected by gravity).

**Figure 13 sensors-23-07236-f013:**
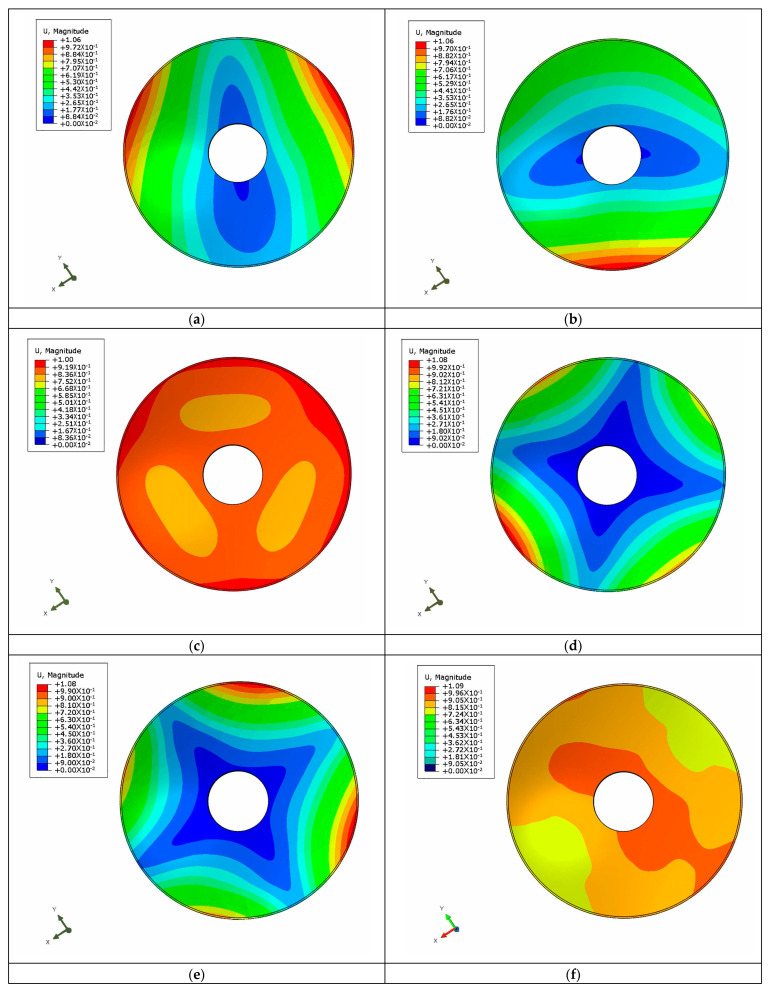
Cloud picture of the first six order modes of the primary mirror. (**a**) First-order mode; (**b**) Second-order mode; (**c**) Third-order mode; (**d**) Fourth-order mode; (**e**) Fifth-order mode; (**f**) Sixth-order mode.

**Figure 14 sensors-23-07236-f014:**
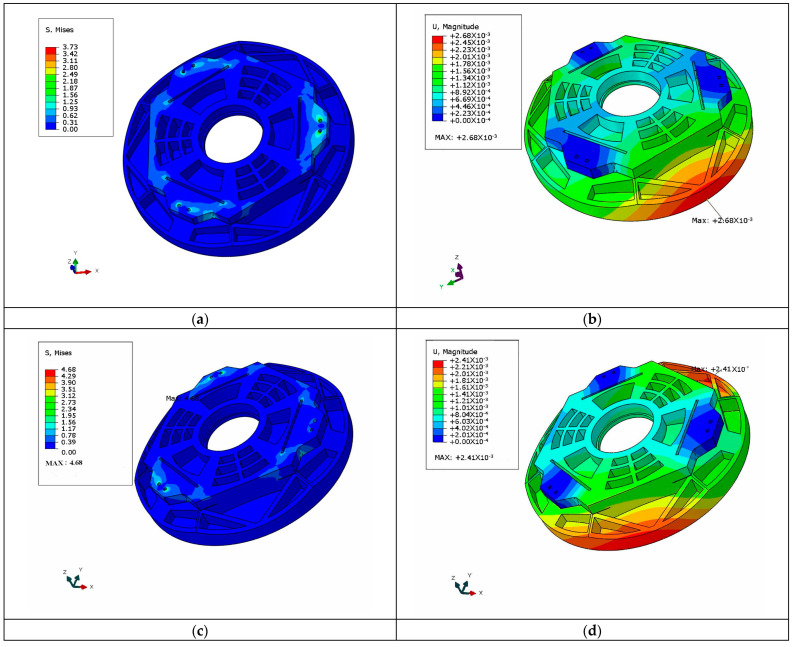
Cloud picture of the primary mirror three-way 20 g overload stress and displacement: (**a**) Cloud picture of X-direction impact stress; (**b**) Cloud picture of impact displacement in X-direction; (**c**) Cloud picture of Y-direction impact stress; (**d**) Cloud picture of impact displacement in Y-direction; (**e**) Cloud picture of Z-direction impact stress; (**f**) Cloud picture of impact displacement in Z-direction.

**Table 1 sensors-23-07236-t001:** Properties and physical properties of commonly used mirror materials.

Materials	Modulus of Elasticity E (GPa)	Density ρ (g·mm^−3^)	Specific Stiffness (E/ρ)	Thermal Conductivity k (m·k)	Coefficient of Linear Expansion α (10^−6^/K)	Temperature Conductivity K/α
Microcrystalline glass	91	2.53	35.97	1.64	0.05	32.80
Silicon	131	2.33	56.22	137	2.60	52.69
ULE	67	2.21	30.31	1.31	0.03	43.67
Beryllium	287	1.85	155.14	216	11.40	18.95
Aluminum (6061-T6)	68.2	2.68	25.18	167	23.6	7.40
Silicon carbide	400	3.20	125.00	155	2.40	64.58

**Table 2 sensors-23-07236-t002:** Predicted results of the support parameter of the mirror to be tested.

Sample to Be Tested		Predicted Results		Sample to Be Tested		Predicted Results	
Support Position	Support Depth	Support Position	Support Depth	Support Position	Support Depth	Support Position	Support Depth
89	69	89	69	73	77	73	77
76	41	76	41	72	65	72	65
84	77	84	77	76	81	76	81
91	57	91	57	96	45	98	45
82	65	82	61	93	73	93	73
93	61	94	61	85	45	84	45
74	53	74	53	98	85	98	85

**Table 3 sensors-23-07236-t003:** Results of mirror’s surrogate model assessment (The original model).

Appraisal Indicators	Support Position (Accurate Forecast/Overall Number)	Support Interval (Exact Forecast/Overall Number)
Appraisal results	9/12	13/14

**Table 4 sensors-23-07236-t004:** Multiple optimizations of the predicted results of the support parameter to be tested.

Sample Serial Number	Sample to Be Tested		Predicted Results		Number of Optimizations
	Support Position	Support Depth	Support Position	Support Depth	
1	89	69	89	69	0
2	76	41	76	41	0
3	84	77	84	77	0
4	91	57	91	57	0
5	82	65	82	65	1
6	93	61	93	61	1
7	74	53	74	53	0
8	73	77	73	77	0
9	72	65	72	65	0
10	76	81	76	81	0
11	96	45	97	45	2
12	93	73	93	73	0
13	85	45	85	45	1
14	98	85	98	85	0

**Table 5 sensors-23-07236-t005:** Results of mirror’s surrogate model assessment.

Appraisal Indicators	Support Position (Accurate Forecast/Overall Number)	Support Interval (Exact Forecast/Overall Number)
Appraisal results	11/12	14/14

**Table 6 sensors-23-07236-t006:** PV and RMS values of the mirror’s optical surface under gravity in the X-, Y- and Z-directions.

Working Conditions	PV Value (nm)	RMS Value (nm)
X-direction by gravity	37.86	7.62
Y-direction by gravity	40.91	7.62
Z-direction by gravity	45.26	10.44

**Table 7 sensors-23-07236-t007:** Results of mirror modal analysis.

Number of Steps	Frequency (HZ)	Vibration Type
First-order	1018.3	The mirror rotates around the Y-axis
Second-order	1019	The mirror rotates around the *X*-axis
Third-order	1109.1	The mirror rotates around the Z-axis
Fourth-order	1647	The mirror edge corner oscillates around the X-axis
Fifth-order	1647.8	The mirror edge oscillates around the Y-axis
Sixth-order	2740.6	Mirror panning around the Y-axis

**Table 8 sensors-23-07236-t008:** Results of mirror overload analysis.

Direction	Maximum Stress/Mpa	Maximum Stress Position	Maximum Displacement/μm	Maximum Displacement Position
X	3.73	Lateral wire-cutting connection	2.68	Panel edge of the primary mirror
Y	4.68	Lateral wire-cutting connection	2.41	Panel edge of the primary mirror
Z	10.98	Lateral wire-cutting connection	4.60	Panel edge of the primary mirror

## Data Availability

The data are contained within the article.

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
