# Peer review of "The Application of a Topology Optimization Algorithm Based on the Kriging Surrogate Model in the Mirror Design and Optimization of an Aerial Camera"

_sensors, 2023, doi:10.3390/s23167236_

Round 1

Reviewer 1 Report

Yubo Zhao et al., proposed an integrated mirror light engine design optimization process is proposed for an aviation optoelectronic device. In my view, the topic is interesting. However, some additional discussion and analysis can be added to make this work more complete. My detailed comments are listed below.

1.      The novelty of the work is not stated properly. Clearly write the novelty.

2.      What about the polarization of the proposed system as it is going to be used for free space communications.

3.      There are few grammatical errors. Rectify them.

4.      Clearly explain lightweight structure of the mirror.

5.      Why author fixed 632.8nm wavelength?

There are few grammatical errors. Rectify them.

Reviewer 2 Report

The authors demonstrate an algorithm-based design optimization for the primary mirror in Cassegrain optical system of aerial camera. They achieved RMS of the primary mirror below 1/40 of the wavelength (at 632 nm) and small construction weight that is mechanically reliable in harsh environment conditions of aerospace camera operation (the latter was verified with finite element modeling involving up to 20g loads). The method is efficient and encouraging physical implementation, I recommend the article for publication subject to addressing few comments/questions.

1) Line 52: in this line term "precision value" is used for the first time, but already as acronym "PV". It will be easier for reader if written like "...minimum precision value (PV) of the mirror.. "

2) Line 98: authors mention mirror material is an aluminium alloy, but do not specify any details. If this is one of standard Al alloys (for example, 7075 Al-Zn alloy), please indicate in text and possibly add physical properties to Table 1.

3) Line 115: in the sentence part "Mirror optical surface surface shape precision..." one word "precision" seem to be enough

4) Line 119: does calculation in empirical formula (1) include an aperture of the primary mirror?

5) Line 166: Figure 2 has a typo: while text discusses design and non-design domains of the finite element model, both domains are named "non-designed" in the figure

6) Line 200: Figure 3 lacks color bars and explanation which parameter was optimized

7) Line 359: authors discuss that addition of original points to areas with highest MSE greatly reduces errors of surrogate model (which is demonstrated in Figures 7 and 8), but at the expense of longer computation time. Could you please specify how big is a relative increase in computation time? Model precision seems great after this iteration.

8) Line 488: Figure 10 lacks color bar with unit explanation. Or is it just an image showing optimized geometry of the mirror?

9) General question: authors report simulated RMS as low as 7.62 nm, what would be a cost-efficient way to manufacture mirror with such surface quality?

Minor changes needed in structures of some sentences. For example, had to read sentences in lines 232-234 few times until I understood.
